# Thermally Stable Nitrothiacalixarene Chromophores: Conformational Study and Aggregation Behavior

**DOI:** 10.3390/ijms21186916

**Published:** 2020-09-21

**Authors:** Anton Muravev, Tatiana Gerasimova, Robert Fayzullin, Olga Babaeva, Ildar Rizvanov, Ayrat Khamatgalimov, Marsil Kadirov, Sergey Katsyuba, Igor Litvinov, Shamil Latypov, Svetlana Solovieva, Igor Antipin

**Affiliations:** 1Arbuzov Institute of Organic and Physical Chemistry, FRC Kazan Scientific Center, Russian Academy of Sciences, Arbuzov str. 8, 420088 Kazan, Russia; gryaznovat@iopc.ru (T.G.); fayzullin@iopc.ru (R.F.); olbazanova@iopc.ru (O.B.); rizvanov@iopc.ru (I.R.); ayrat_kh@iopc.ru (A.K.); kamaka59@gmail.com (M.K.); katsyuba@iopc.ru (S.K.); litvinov@iopc.ru (I.L.); lsk@iopc.ru (S.L.); svsol@iopc.ru (S.S.); 2Butlerov Institute of Chemistry, Kazan Federal University, Kremlevskaya str. 18, 420008 Kazan, Russia; iantipin54@yandex.ru

**Keywords:** nitrothiacalix[4]arenes, Mitsunobu alkylation, *1,2-alternate* stereoisomer, TG/DSC analysis, X-ray diffractometry, DFT study, Langmuir monolayers, reflection-absorption spectroscopy

## Abstract

Achieving high thermal stability and control of supramolecular organization of functional dyes in sensors and nonlinear optics remains a demanding task. This study was aimed at the evaluation of thermal behavior and Langmuir monolayer characteristics of topologically varied nitrothiacalixarene multichromophores and phenol monomers. A nitration/azo coupling alkylation synthetic route towards partially O-substituted nitrothiacalixarenes and 4-nitrophenylazo-thiacalixarenes was proposed and realized. Nuclear magnetic resonance (NMR) spectroscopy and X-ray diffractometry of disubstituted nitrothiacalix[4]arene revealed a rare *1,2-alternate* conformation. A synchronous thermal analysis indicated higher decomposition temperatures of nitrothiacalixarene macrocycles as compared with monomers. Through surface pressure/potential-molecular area measurements, nitrothiacalixarenes were shown to form Langmuir monolayers at the air–water interface and, through atomic force microscopy (AFM) technique, Langmuir–Blodgett (LB) films on solid substrates. Reflection-absorption spectroscopy of monolayers and electronic absorption spectroscopy of LB films of nitrothiacalixarenes recorded a red-shifted band (290 nm) with a transition from chloroform, indicative of solvatochromism. Additionally, shoulder band at 360 nm was attributed to aggregation and supported by gas-phase density functional theory (DFT) calculations and dynamic light scattering (DLS) analysis in chloroform–methanol solvent in the case of monoalkylated calixarene **3**. Excellent thermal stability and monolayer formation of nitrothiacalixarenes suggest their potential as functional dyes.

## 1. Introduction

Control of supramolecular interactions plays an increasing role in the performance of organic dyes [1] and macrocycles [2] in a variety of applications including nonlinear optical (NLO) materials, sensors, drug carriers, catalysts, and explosives. Among macrocycles, calix[4]arenes should be particularly mentioned, whose facile stereo-, regio-, and iteroselective modifications [3] provide diverse architectures displaying multipoint binding [4], allosteric regulation [5], and self-assembly behavior [6,7]. Calixarenes containing nitro groups on the upper rim offer potential applications as NLO materials [8], redox probes [9], and colorimetric chemosensors [10] due to the formation of a push-pull multichromophoric system consisting of a nitro group as the electron-withdrawing unit, a benzene ring of calixarene as the aromatic spacer, and a lower rim alkoxy group as the electron donor. Ultrathin films based on calixarene chromophores (up to few monolayers) are particularly attractive objects due to their accessibility of receptor units, as well as highly organized arrangement of molecules (along with preorganized chromophoric fragments within calixarene molecule), and a high packing density for sensor and NLO applications, respectively. However, only two examples of monolayer formation by nitrocalixarene chromophores (both by Langmuir technique) have been reported so far [11,12].

Heat stability is another factor regarding efficiency of dyes and explosives and calixarenes were reported to be heat stable [13]. In spite of the fact that nitrocalixarenes have higher melting points than monomer phenols, their thermal characteristics have not been explored in detail.

There are two common topologies of calixarene scaffold with electron-withdrawing nitro groups on the upper rim (Figure 1), tetranitrocalixarenes with symmetrically substituted lower rim (type I) [14,15,16] and partially upper rim substituted calixarenes (type II) [17,18,19]. Because partially lower rim-substituted calixarenes are key intermediates for the design of heterotopic ligands acting as switchable or dual-mode sensors, the development of a convenient method for their synthesis is highly desirable. Unfortunately, there are few literature examples of partially O-alkylated tetranitrocalix[4]arenes (type III) [20,21,22,23]. They have shown mixed conformational behavior as compared with donor-substituted calixarenes on the upper rim as exemplified by the unusual *1,3-alternate* conformation of diallyloxy nitrothiacalix[4]arene in solid state [20] and a *partial cone* conformation of mono(ethoxycarbonyl)methoxy calix[4]arene in solution [23]. To date, type III ligands have only been studied as metal ion receptors in solution, whereas physicochemical and other functional characteristics have not been reported. Thus, the lack of reports on the synthesis, conformational behavior, and properties of type III nitrocalixarenes hampers their further exploitation as advanced materials and leaves aside the effect of lower rim substitution pattern on functional characteristics.

As part of our ongoing studies on the synthesis and self-assembly of heterotopic thiacalixarenes, we synthesized a series of type I–III nitrothiacalixarene chromophores and their monomer phenolic structural counterparts and evaluated their thermal stability and self-organization in solution, solid state, and ultrathin films.

## 2. Results

### 2.1. Design of the Chromophores

Figure 2 outlines structural variations of type I–III nitrothiacalixarene scaffold and phenolic monomer in order to assess their effect on thermal stability and aggregation behavior in ultrathin films. The following factors were considered: nature of spacer fragment, through a comparison of nitrothiacalixarenes and 4-nitrophenylazo-thiacalixarenes; number of alkyl fragments on the lower rim of calixarene (zero, one, and two); number of nitro groups on the upper rim of macrocycle (two and four); and cluster effect, through a comparison of thiacalixarene and model phenolic chromophores.

### 2.2. Synthesis of the Chromophores

Type I calixarenes with hydroxyl groups at the lower rim and nitro and 4-nitrophenylazo groups at the upper rim were prepared using reported literature procedures (compounds **1** and **4**) [24,25] and were employed as precursors for subsequent modification into type III ligands (Figure 1).

Electrophilic substitution of thiacalix[4]arene at the upper rim followed by lower rim functionalization with halides using alkali metal carbonates or with alcohols in triphenylphosphine (TPP)/diethyl azodicarboxylate (DEAD) system is a common strategy to decorate the scaffold. However, electron-withdrawing substituents decrease nucleophilicity of phenolate-anion and require higher temperatures and longer reaction time to access type III structures. Due to the base-promoted ether-group cleavage in distally disubstituted thiacalixarenes, which is enhanced at elevated temperatures and longer reaction times [26,27], nitrocalixarene **1** was alkylated with 1-dodecanol and 1-octanol (synthesis, monolayer formation, and optical characteristics of the product **2b** was published as a preliminary result in [28]) (Scheme 1). Interestingly, the extent of lower rim substitution could be controlled by reaction temperature. A two-day stirring at 40 °C afforded disubstituted products **2a**,**b** with 46–69% yields, while a five-day stirring at 25 °C gave mono iteromer **3** with 40% yield at low conversion of nitrocalixarene **1**. This is the first example of monosubstituted calixarene accessed by the Mitsunobu reaction. Similarly, azocalixarene **4** was converted at 40 °C to distally disubstituted derivative **5** in 29% yield, whereas stirring at 25 °C did not produce mono iteromer.

To extend a series of thiacalixarene chromophores to type II structures, thiacalix[4]arene **6** was converted into distally disubstituted intermediate **7** under Mitsunobu conditions at 2.2-fold excess of TPP/DEAD system (Scheme 2). It should be noted that an increase in the TPP/DEAD ratio from 2.2 to 4 equiv. also afforded a trisubstituted product **7′**. Selective nitration of the derivative **7** using the mixture of nitric and acetic acids, at various temperatures, failed due to the oxidation of bridging sulfur atoms and partial dealkylation of either groups (as evidenced by mass-spectrometry data). Under milder conditions, using nitrosonium nitrate stabilized by 18-crown-6, product **8** was isolated in 24% yield. Similarly, trisubstituted product **7′** was mononitrated at phenolic ring to give product **8′**. Unfortunately, analogous type II disubstituted azo derivative was not isolated as an individual product after column chromatography due to the impurity of monoazothiacalix[4]arene and full conversion of this impurity into target product was not achieved even after a one-week heating of azothiacalixarene **4** at 60 °C with an eight-fold excess of diazonium salt. These results show that only nitration/azo coupling alkylation sequence is effective for synthesis of type III nitrocalixarene chromophores, whereas inverse functionalization sequence results only in partial upper rim substituted type II chromophores.

Finally, monomer counterparts **10** and **12** were synthesized in predictably high yields through the reaction of octyl bromide with corresponding phenols **9** and **11** (Scheme 3).

### 2.3. Conformational Study of the Chromophores

All the synthesized compounds were fully characterized using spectroscopic and spectrometric techniques (Appendix A). Having mentioned unexpected stereoisomeric forms of type III nitrocalix[4]arenes in the Introduction [20,23], conformational behavior of the chromophores under study is not a priori apparent. The ^1^H nuclear magnetic resonance (NMR) spectra of nitro derivatives **3** and **2** and nitrophenylazo product **5** in CDCl_3_ are typical of mono- and disubstituted iteromers on the thiacalixarene platform showing AB quadruplet and two singlets at 8.7–8.4 ppm in the case of compound **3** and, in the case of compounds **2a** and **5**, a pair of singlets (Figure 3). High symmetry of the spectrum of compound **5** and a large chemical shift (CS) difference between the singlets of phenolic protons (ca. 8.5 and 7.7 ppm) suggest that azocalixarene **5** exists as a *cone* conformer in solution. In analogy to other distally disubstituted thiacalixarenes, a *cone* stereoisomeric form was initially suggested for nitrothiacalixarene **2b** due to high symmetry of the spectrum [28]. However, close arrangement of two singlets of aromatic protons (8.50 and 8.49 ppm) could also indicate fast equilibrium of *1,2-alternate* and *partial cone* forms due to through-the-annulus rotation of OH-groups (Appendix A) [29]. In addition, high-field chemical shifts (CSs) of methylene groups (from 4.5 to 4.0 ppm and from 1.2 to 0.9–1.1 ppm, Appendix A) of compound **2** suggests the influence of magnetic anisotropy of proximal inverted aromatic rings. The absence of such high-field CSs in the case of compound **3** confirms that it adopts a *cone* rather than a *partial cone* conformation. Again, the conformation of type II chromophores could be deduced from CS of oxymethylene protons, which takes the value of 4.33 ppm in the case of dinitro product **8** and corresponds to a *cone* conformation. NMR spectrum of mononitro product **8′** shows triplet at 6.97 (1H) and a doublet-of-doublets at 6.42 ppm (2H) corresponding to *para* protons and a doublet (7.57 ppm) and few multiplets of *meta* protons at 7.6–6.5 ppm (Appendix A). The retention of the *partial cone* stereoisomeric form in compound **8′** was deduced from the same CSs of oxymethylene protons (4.15 and 3.94 ppm) as those in compound **7′**.

The *anti* arrangement of dodecyl groups in compounds **2** is highly likely due to the equivalent number of H-bonds in both a *partial cone* and *1,2-alternate* conformations in the equilibrium mixture (two H-bonds), whereas syn oriented alkyl chains would result in two less stable equivalent *partial cone* conformers with one H-bond. The final evidence of the *anti* arrangement of dodecyl groups in compound **2a** was gained by X-ray diffraction data. Compound **2a** crystallizes in the triclinic space group $P\bar{1}$ with one half of a molecule in asymmetric cell, thus adopting *1,2-alternate* conformation (Figure 4). Such a conformation of thiacalixarene **2a** in the crystals is unique among distally disubstituted calixarenes with large substituents (more than three carbon atoms). Although distally disubstituted calixarenes with small groups on the lower rim could exist in solution as a *cone* and *1,2-alternate* equilibrium mixture, only crystal structure of *1,2-alternate* dipropoxy derivative has been reported so far [30].

The conformation of molecule **2a** in the crystals is stabilized by two intramolecular hydrogen bonds O(1A)–H(1A)∙∙∙O(1B) between hydroxyl groups and oxygen atoms of the alkoxy substituents. The crystal structure is formed by weak intermolecular interactions and characterized by quite high packing density with the Kitaigorodsky index (percentage of the occupied space in the unit cell) of 69.3%.

The density functional theory (DFT) calculations on simplified models of chromophores **5** and **2a** (with hexyl chain instead of dodecyl, **5′** and **2a’**) were carried out to evaluate energy differences of their main forms (*pinched cone* (*PC*), *distorted cone* (*DC*), *1,3-alternate*, *1,2-alternate*, and *partial cone*) (Appendix A). The calculations showed that *PC* stereoisomer was the most favorable conformation in the case of **5′**, whose energy was only 0.8 kcal/mol lower than that of *1,2-alternate* form, whereas they were nearly identical in energy in the case of **2a’**. Given the similar electron-withdrawing effects of nitro and nitrophenylazo groups at the upper rim of thiacalixarene platform and similar energies of *PC* and *1,2-alternate* stereoisomeric forms, therefore, it is unclear why disubstituted iteromers **2a** and **2b** adopt *1,2-alternate* conformation with *anti* arranged substituents, whereas product **5** adopts a *cone* conformation with *syn*-oriented dodecyl groups. One hypothesis is that calixarene precursors **1** and **4** exist in different stereoisomeric forms in solution, which affects the energy barrier of conversion into disubstituted products. In spite of identical orientations of precursors **1** and **4** in solid state (P-1 space group, *1,2-alternate* conformation) [31,32], there were different crystallization conditions, and an unambiguous statement on the true stereoisomeric form of calixarenes **1** and **4** in solution could not be made from these data. A conformational study of nitro-containing thiacalixarene precursors in solution was also restricted by their low solubility in common deuterated solvents including those used for the Mitsunobu reaction (toluene, tetrahydrofuran). Thus, thermodynamic or kinetic control of the Mitsunobu reaction of nitrothiacalixarenes **1** and **4** with dodecyl alcohol giving compounds **2** and **5** could not be determined and this hypothesis could not be verified.

Another hypothesis is that the OH group distal to the alkyl chain can rotate through the thiacalixarene rim in the case of monosubstituted **3** upon Mitsunobu alkylation and forms an H-bond with the nitro group, thus, stabilizing the *anti* arrangement of the proximal OH groups in the intermediate, whereas such stabilization is impossible in azo derivative **4** due to the larger distance from the OH group to the NO_2_ group.

### 2.4. Thermal Stability Study

A synchronous thermal gravimetry and differential scanning calorimetry (TG/DSC) analysis was carried out in the temperature range of 25–550 °C to evaluate thermal stability of the chromophores (Table 1, corresponding curves are given in Appendix A). It should be noted that thermal decomposition data of type I chromophores **1** and **4** differ from previously reported values (*T*_d_ (DSC) = 280 °C (compound **1**) [31] and 210 °C (compound **4**) [33]), presumably due to a different synthetic route to compound **1** (glacial acetic acid/nitric acid in this work and KNO_3_/AlCl_3_ in [31]) and isolation of compound **4** (washing with HCl in this work and recrystallization from pyridine in [33] (see [9] therein)) (Appendix A). In this work, compound **1** decomposed at 366.0 °C with an exo peak of 730.9 J/g corresponding to ~39% weight loss. Azothiacalix[4]arene **4** shows an exo-peak of 809.2 J/g at 372.9 °C coinciding with onset of decomposition according to the thermal gravimetry(TG) curve. These corrected values indicate that thermal stability of type I chromophores is higher than those of well-known hexanitrostilbene (HNS) and 2,6-bis(picrylamino)-3,5-dinitropyridine (PYX) explosives and comparable to perspective octanitrooxacalixarene explosive [34] and could be interesting for heat-resistant explosion studies. Thermal stability decreases with a transition from multidipolar thiacalix[4]arenes to model phenols **9** and **11**. Their TG curves indicate decomposition at 181 °C and 262 °C, respectively.

Type II chromophore **8**, with two upper rim NO_2_ groups and two lower rim C_12_ chains, decomposes at lower temperatures (≈300 °C according to TG) and DSC exo-peak appears at 315.6 °C. A further decrease in thermal stability of thiacalixarenes is recorded in type III chromophores. Replacement of C_12_ chain in compound **2a** by octyl (**2b**) marginally alters heat stability (DSC peak at ca. 290 °C). Surprisingly, nitrothiacalixarene **3** with one dodecyl group displays a lower temperature corresponding to the DSC peak (274 °C), in spite of a more energetically favorable *cone* conformation (**3**) and lower number of alkyl chains acting as destabilizing factor in compounds **2a**,**b**. Thermal stability of type III chromophore **5** with azo groups is comparable to those of nitrophenylazophenols **11** and **12** (corresponding DSC decomposition peaks at 298, 292, and 335 °C).

Thus, the thermal analysis shows that types I–III nitrocalixarenes are generally more thermally stable than phenols, although there are no correlations among thermal stability and the number of nitro/alkyl groups and the nature of aromatic spacer in calixarene, presumably due to different stereoisomeric forms of the chromophores and possible conformational equilibria at high temperatures.

### 2.5. Langmuir Film Formation

The Langmuir method was used in this work as a universal approach to monolayer formation with the unique ability to control packing density of the molecules through surface pressure parameter.

The nitrophenols **9** and **11** did not form monolayers due to their solubility in water subphase (0.08 M and 1.6 × 10^−5^ M, respectively). Furthermore, surface pressure at the air–water interface did not increase after spreading their solutions in chloroform over the subphase and movement of barriers, which indicated no adsorbed film formation by these ligands. Although alkylated product **10** formed monolayer after spreading 8 × 10^−4^ M solution in CHCl_3_ on water subphase, its low collapse pressure (<5 mN/m) showed that it could not form rigid films on solid substrate.

The formation of Langmuir–Blodgett (LB) films by the macrocycles was firstly optimized on the example of compound **2b** [28]. To form a monolayer, 150 μL of 10^−4^ M solution of compound **2b** in chloroform was spread onto the water surface with an area of nearly 240 cm^2^, which produced in the starting molecular area of nearly 200 Å^2^. Unfortunately, compound **2b** did not form stable monolayers in the described conditions. The analysis of compression isotherms (dependence of the surface pressure in monolayer on the area per molecule of compound **2b** on the water subphase) showed that limiting molecular areas were less than 80 Å^2^, which proved the impossibility of the formation of a true monolayer in the system and was presumably related to the aggregation of the molecules of **2b** in solution, as recorded by dynamic light scattering (Table 2). To break the aggregate structure, a five-fold dilution of calixarene solutions in chloroform (from 10^−4^ to 2 × 10^−5^ M) was attempted; however, this resulted in a gradual increase in the size (from 10 to 100 nm according to the number-averaged particle size distribution (PSD)) (Table 2).

Assuming that, in nonpolar chloroform, inverse micelles of compound **2b** with the orientation of nitro groups inwards of the micelle could be formed, methanol was added to chloroform to increase the polarity of the solution and to decrease the thermodynamic stability of the mentioned aggregates. It was revealed that aggregates disappeared at 15% content of methanol in the mixture (Table 2). The absence of aggregates in the solution provided true Langmuir monolayer formation in the case of calixarene **2b** (Figure 5a) with the mean molecular area of *A*_0_ ≈ 400 Å^2^/molecule and compressibility modulus of $C_{s}^{–1}$ = 26 mN/m at the first linear section of the isotherm, which corresponded to the liquid-expanded state of the monolayer at this section [35]. Surface potential-molecular area isotherms, which were recorded using a vibrating plate method simultaneously with surface pressure-molecular area isotherms, demonstrated low surface potential (−10 to −30 mV) over the monolayer compression due to near-zero dipole moment of *1,2-alternate* stereoisomer **2b**. Such invariance of surface potential values indicated the constant orientation of nitrocalixarene molecules. The monolayers of compound **2b** were successfully transferred onto mica substrate with a high transfer ratio (TR). A strong tendency of compound **2b** towards aggregation was shown using atomic force microscopy (AFM) technique (Figure 5b). A variety of aggregates with the sizes of ≈50 and 500 nm was observed in AFM images of the films on the mica substrate.

Unlike calixarene **2b**, nitrocalixarenes **3**, **5**, and **8** did not form nanoaggregates in chloroform at 10^−4^ to 10^−5^ M and were spread on the water subphase from chloroform solutions. Subsequent lateral compression at the air–water interface resulted in the increase in surface pressure starting from the molecular areas of about 90 Å^2^ (**3**), 105 Å^2^ (**5**), and 120 Å^2^ (**8**), which were close to theoretical values of vertically oriented thiacalixarenes with alkyl fragments. In the case of compounds **1** (≈45 Å^2^) and **4** (≈35 Å^2^), which were spread from chloroform–methanol solutions due to low solubility in CHCl_3_, the limiting molecular areas were twice as small as those of vertically oriented molecules (Figure 6). In the two latter cases (type I chromophores), either bilayer was formed or the compounds were partially dissolved in water subphase. The surface pressure values of calixarene monolayers indicated that the most rigid monolayers were formed from monoalkylated calixarene **3** (collapse pressure was 40 mN/m), which also displayed a comparatively large compressibility value ($C_{s}^{–1}$ = 120 mN/m) (Figure 6 and Table 3).

The value of surface potential difference (Δ*V*) below and above the monolayer was directly proportional to the total dipole moment of the molecules in monolayer and could provide further evidence of their orientation. Surface potential measurements of the calixarene monolayers spread on the water subphase support stereoisomeric forms of type II and III calixarenes, which are discussed in the Characterization section in Materials and Methods. The lowest absolute values of Δ*V* correspond to *1,2-alternate* stereoisomer **2b** (Table 3) and change marginally upon monolayer compression (Figure 5a). On the contrary, the *cone* stereoisomers **3**, **5**, and **8** display higher surface potentials (Δ*V* = 125–230 mV), which are caused by similar orientation of D–π–A dipoles in calixarene chromophore (Figure 6c–e). There is a positive trend of the increase in Δ*V* in the case of compounds **3** and **5**, which can be caused by a gradual change from C_2_ symmetry of calixarenes to C_4v_ possessing higher dipole moment with an increase in surface pressure. In the case of compound **8**, a C_4v_ symmetry of the molecule before compression of monolayer can be suggested due to a different H-bonding pattern. A large negative surface potential of monosubstituted iteromer **3** can be rationalized by a different orientation of calixarene molecules at the air–water interface, with hydroxyl groups immersed to water subphase and alkyl chain spread in parallel, whereas nitrocalixarenes **5** and **8** with hindered OH groups could interact with water molecules through NO_2_ groups. This suggestion is supported by a much higher collapse pressure of chromophores **1**, **3**, and **4** with OH groups that are accessible for water molecules. Deprotonation of OH group could also explain the negative surface potential of compound **3** (pK_a,2_ value of nitrothiacalixarene **1** is 6.48 [36]). In the case of type I nitrocalixarene **1**, a gradual increase in the absolute value of surface potential upon film compression to 100 Å^2^ indicates a change of molecular conformation possessing higher dipole moment than that of *1,2-* or *1,3-alternate* (Figure 6a) and presumably a *cone* conformer is formed at the air–water interface. An analogous, although positive, increase in surface potential recorded in the presence of azo chromophore **4** (Figure 6b) could be caused by transformation into a *cone* stereoisomer without deprotonation of OH groups, resulting in lower surface pressure after compression.

To gain further evidence for the effect of hydrogen bonding interactions on monolayer stability of nitrocalixarenes **1**–**3** and **8**, the transfer pressure was maintained constant and the change in the barrier position was recorded over time. Figure 7 shows the dependence of the change of barrier position over time before transfer to the solid substrate and indicates that the monolayers of **1** and **3** are the most stable, with the smallest change of barrier position, in contrast to compounds **2b** and **8**, where hydroxyl groups are hindered by the alkyl chain.

One to five layers of the nitrothiacalixarenes **1**, **2b**, **3**, and **8** were successively transferred to indium-tin oxide (ITO) substrate, with a transfer ratio close to 1, except for nitrothiacalixarene **1** (TR ≈ 0.5). Film surface morphology was investigated by AFM (Figure 8, Table 4). In the case of nitrothiacalix[4]arene **1**, only one layer was recorded (Figure 8b), which gave a diffuse image due to the instability of the layer and its low adhesion to the substrate. The surface with one monolayer of compound **8** (Figure 8f) possessed a roughness of nearly 3 nm with a mean thickness of 1.5 nm, whereas five monolayers (Figure 8g) appeared as a hummocky surface (mean thickness of 12 nm) comprised of irregular-shaped polydisperse particles with heights in the range from 15 to 39 nm and half-height diameters of 72 to 382 nm. The surface with one monolayer of **2b** (Figure 8c) represented the film with large (heights, on average, of 40 nm and half-height diameters, up to 450 nm) irregular-shaped particles, with a mean thickness of 1.2 nm. The most homogeneous surface was observed in the case of one and five monolayers of compound **3**, whose roughness and surface corresponded to about 1 and 2 nm for each monolayer, respectively (Figure 8d,e). The surface roughness of the ITO substrate (Figure 8a) complicates the interpretation of the morphology of nitrothiacalixarene monolayers; however, it is clear from the data that effective transfer of the monolayer, as well as their uniformness in the film are determined by structural factors, such as the presence of long-chain alkyl fragments on the lower rim of calixarenes. Thus, compound **3** with one dodecyl fragment is the optimal structure for homogeneous film formation.

Langmuir monolayer experiments show that type **I** nitrocalixarene chromophores could not form true monolayers in spite of the presence of four hydrophilic OH groups, which could be caused by partial dissolution of these ligands in the subphase or bilayer formation (the behavior already mentioned for the parent *tert*-butyl-thiacalixarene [37]), and give an ill-defined LB film. When there are dodecyl units on the lower rim of macrocycle, monolayer formation ability is enhanced and true monolayers are formed for type II and III chromophores, which are effectively transferred onto the solid substrate. Phenolic monomers, although capable of assembly into monolayers at the air–water interface, possess very low collapse pressures and do not form rigid LB films.

### 2.6. Spectrophotometric Study of Nitro Chromophores in Solution, Air–Water Interface, and on Solid Substrate

The surface pressure/potential-molecular area and monolayer stability measurements, as well as successful LB transfer to solid substrates showed the potential of nitrothiacalixarenes in ultrathin film formation. Nevertheless, orientation of these multidipolar chromophores in the monolayer could not be clearly deduced from the Langmuir experiments, and required additional studies. To shed light on their orientation, optical characteristics of the monolayers were investigated using in situ UV/Vis reflection-absorption spectroscopy (UVRAS) and compared to electronic absorption spectra of corresponding solutions of the compounds in spreading solvents (Appendix A).

In the case of calixarene **2b** at the air–water interface, a bathochromic shift of the adsorption band was observed upon monolayer formation and compression (λ_max_ = 292 nm) (Figure 9) with respect to the chloroform solution (λ_max_ = 274 nm), which indicated the possibility of solvatochromism. This suggestion was evidenced from UV spectra in CHCl_3_ and chloroform–MeOH solvent (Figure 9, inset), i.e., modification of solvation shell of calixarene upon addition of 15% methanol (*v*/*v*) resulted in bathochromic shift of the main absorption band in UV/Vis spectrum by 3 nm [28].

In analogy to nitrothiacalix[4]arene **2b**, a bathochromic shift of absorption bands of calixarene molecules was observed in all monolayers on the water subphase, which took the values from 6 nm in the case of compound **8** to 44 nm in the case of nitrocalixarene **1** (Table 5). Such a high shift in the case of compound **1** could be caused by its low pK_a_ values in water (pK_a,1_ = 2.75 and pK_a,2_ = 6.48) [36], which resulted in broadening of absorption band with a transition from chloroform to air–water interface due to the presence of deprotonated forms (Appendix A). Considerations of pK_a_ values could also apply to rationalization of a bathochromic shift of monosubstituted product **3**.

To gain a deeper insight into the aggregation behavior of monoalkoxy derivative **3**, with the largest surface potential among the types I–III chromophores, its Langmuir monolayers were transferred onto quartz substrate, which did not absorb UV light at 400 nm as in the case of ITO substrate, and its spectrophotometric behavior was studied in detail. Apart from the band at 273 nm, the electronic absorption spectra of the LB films consisting of one and five monolayers of calixarene **3** on quartz displayed a shoulder band at 365 nm (Figure 10a). It should be noted that the same shoulder appeared in the calixarene monolayer at the air–water interface upon lateral compression, as well as in monolayers of other calixarenes.

In order to evaluate the effect of the environment with the transition of chromophore from CHCl_3_ to water subphase, up to 100% of MeOH was added to the CHCl_3_ solution of nitrocalixarene **3**. In analogy to compound **2b**, absorption band maximum at 273 gradually shifted to 278 nm. However, a shoulder at 350 nm was also recorded and its intensity increased upon addition of MeOH cosolvent (Figure 10b). Thus, changes of electronic absorption spectra of nitrothiacalixarenes with a transfer from solution to air–water interface and further to LB films reveal the following two trends: (1) a bathochromic shift of absorption band maximum, which is contributed to by solvatochromism and acid–base equilibrium and (2) an increase in the intensity of the shoulder band at higher wavelength governed by a change of solvent polarity.

### 2.7. Contribution of Aggregation Phenomena to Optical Absorbance Characteristics of Nitrothiacalixarenes

Although the bathochromic shift of absorption maxima at ~280 nm (for compounds **1**–**3** and **8**) and at ~360 nm (for compounds **4** and **5**) was associated with solvatochromic effects, the appearance of additional shoulder in LB films and in solutions with MeOH could not be fully rationalized by solvatochromism and other factors should be considered. One of the possible explanations could be related to aggregation. Indeed, lower solubility of nitrothiacalixarene chromophores in methanol could result in aggregate formation at a high methanol content.

The dynamic light scattering (DLS) experiments detected 207 ± 12 nm aggregates of compound **3** in particle size distribution plot with the change of solvent from chloroform to chloroform–methanol (1:1 *v*/*v*) (Figure 11). This observation confirms the contribution of aggregation phenomena to the appearance of shoulder at 350 nm in UV/Vis spectra. The aggregation ability of calixarene **3** was also demonstrated by powder X-ray diffractogram (PXRD), which showed the most intense reflections at 2θ = 3.44° and 32.48° (strongest) and indicated high crystallinity of nitro derivative **3** (Appendix A).

To further support the effect of aggregation on optical characteristics, quantum chemical calculations of UV/Vis spectra were carried out. Because there are no crystal structures of partially O-alkylated nitrocalix[4]arenes in a cone conformation, several energetically most favorable conformations (*pinched cone*, *partial cone*, and two *distorted cone* forms) of nitrocalixarene **3** were firstly considered in the context of their possible influence on UV/Vis absorption spectra. Nevertheless, all conformations showed similar theoretical spectra with absorption bands at 250 and 290–300 nm. These values of absorption band maxima indicated that the slight shift of absorption band maxima of nitrocalixarenes from 270–280 nm to 280–300 nm on transition from solution to air–water interface and films could be caused by change of conformations of macrocycles along with solvatochromism. As expected, the shoulder at 360 nm could not be rationalized at this stage (Appendix A).

Thus, contribution of aggregation phenomena to UV/Vis spectrum of nitrocalixarene **3** was modeled by considering its dimers in order to discover the above-mentioned shoulder band. Several possible dimer models with different orientation of monomers were optimized as follows: head-to-head dimers with *pinched cone* (**a**) and *distorted cone* (two vertically oriented molecules (**b**) and one vertically and one horizontally oriented molecule (**c**) nitrocalixarene conformers and lateral dimers with parallel (**d**) and antiparallel (**e**) orientation of molecules (Figure 12). Gratifyingly, computations predict an additional low energy band for the head-to-head dimers **a**, **b**, and **c**; this effect is particularly clear in the case of the model **b**. For these dimers, computations show that the longest wavelength absorption bands are associated with electron transitions from the frontier orbitals located at nitrophenol moieties of different molecules, and thus related to intermolecular charge transfer (Appendix A). In the models **d** and **e**, with a larger distance between nitrophenol units of two molecules, low-energy transitions are predicted at shorter wavelengths and associated with intramolecular energy transfer (Appendix A).

Finally, UV/Vis spectra were calculated for tightly packed Langmuir monolayers by modeling parallel oriented calixarene molecules without their optimization (Figure 13). Again, an increase in the number of molecules in the monomer, dimer, and trimer order leads to the appearance and subsequent bathochromic shift of the low-energy band (Figure 13). Thus, the computations predict that aggregation with closely located nitro head groups from different molecules would result in a low-energy band.

The results of computational and experimental studies of UV/Vis absorption spectra and Langmuir monolayer experiments show that nitrothiacalix[4]arenes with dodecyl units on the lower rim could form Langmuir monolayers with oriented dipoles, which indicates noncentrosymmetric aggregation in the case of the *cone* stereoisomers **3** and **5**, as well as parent nitrothiacalix[4]arenes 1 and 4. However, an appealing idea of preorganization of dipoles in nitrothiacalixarene multichromophoric system and oriented Langmuir monolayers for nonlinear optical application was not evidenced by second harmonic generation (SHG) experiments. On irradiation of one to five monolayers of nitrothiacalixarenes **1**–**5** with a 1064 nm laser, the band at 532 nm corresponding to SHG was not recorded. This result can be rationalized by the dipole moment compensation mechanisms in LB films, through the head-to-head or tail-to-tail orientation of calixarene molecules, which is thermodynamically more favorable than head-to-tail orientation. Thus, NLO effects of Langmuir monolayers of nitrothiacalixarene should be investigated with a preliminary exposure of the films to corona discharge and are currently under study.

## 3. Materials and Methods

The solvents were purified by known procedures [38]; the reagents were used as received. All reactions were carried out in argon atmosphere. *p*-Nitrophenol **9** (Alfa Aesar, 99%) was used without further purification, while known procedures were employed to synthesize (*E*)-4-((4-nitrophenyl)diazenyl)phenol **11** [39], thiacalix[4]arene **6** [40], *p*-nitrothiacalix[4]arene **1** [24], *p*-(4-nitrophenylazo)thiacalix[4]arene **4** [25], and ${\mathrm{NO}}^{+}{\mathrm{NO}}_{3}^{–}$*18-crown-6 complex [31].

All NMR experiments were performed with Bruker AVANCE-400/600 MHz (399.9/600.13 MHz for ^1^H NMR and 100.6/150.9 MHz for ^13^C NMR) spectrometers equipped with a 5 mm diameter gradient inverse broad band probe head and a pulsed gradient unit capable of producing magnetic field pulse gradients in the *z* direction of 53.5 G·cm^−1^. The NMR spectra were carried out at 303 K. Chemical shifts are reported in the *δ* (ppm) scale relative to the ^1^H (7.26 ppm) and ^13^C (77.2 ppm) signals of CDCl_3_. Spin-spin coupling constants are given in Hz. Infrared (IR) spectra were recorded in KBr pellets using a Bruker Vector 22 spectrometer in the wavelength range of 400–4000 cm^−1^.

Matrix-assisted laser desorption/ionization (MALDI) mass spectra were recorded on an UltraFlex III TOF/TOF mass spectrometer (Bruker Daltonik GmbH, Bremen, Germany). An Nd:YAG laser, λ = 266 nm, Reflectron mode (resolution 10,000) was used to record high-resolution mass spectra (HRMS) and to obtain exact masses of molecular ions. The data were processed using FlexAnalysis 3.0 program (Bruker Daltonik GmbH, Germany). MTP AnchorChip^TM^ metallic target was used. As a matrix, *para*-nitroaniline was employed (*p*-NA). To record mass spectra with exact mass values, a mixture consisting of the specimen (1 mg/mL, CHCl_3_) and reference substance was prepared. PEG-300 (0.1 mg/mL, CH_3_CN), PEG-1000 (0.1 mg/mL, CH_3_CN), PEG-1500 (0.1 mg/mL, CH_3_CN), 15-crown-5 (0.01 mg/mL, CH_3_OH), 18-crown-6 (0.01 mg/mL, CH_3_OH), and CsCl (1 mg/mL, H_2_O) were used as calibrants in positive-ion recording mode. P*n* (10 mg/mL, CH_3_CN) and calixarenes **3** (0.1 mg/mL, CHCl_3_) and **8** (0.1 mg/mL, CHCl_3_) were used as calibrants in negative ion recording mode. A total of 0.5 μL of matrix solution (10 mg/mL, CH_3_CN) and 0.5 μL of the mixture was successively transferred to the target and evaporated. Relative mass measurement error is <3 ppm.

The TG-DSC studies were carried out on an STA 449 F3 Jupiter differential scanning calorimeter with a thermogravimetric analyzer by Netzsch Company (temperature range was 50–550 °C, Δ*T* = 10.0 °C/min, resolution of TG was 0.00001%, resolution of DSC was 1 μW) in aluminum crucibles in argon atmosphere. The results of the TG-DSC experiments were processed using the NETZSCH Proteus software package.

The DLS measurements were performed on a Malvern Zetasizer Nano particle size analyzer in PCS1115 glass cuvettes thermostated at 25 °C. At least three independent experiments were recorded for each sample. Data were analyzed using Malvern DTS software.

The presence of crystalline component in the chromophore **3** was studied using PXRD on a MiniFlex 600 diffractometer (Rigaku, Tokyo, Japan) equipped with a high-speed D/teX detector. Measurements were carried out using CuKα radiation (40 kV, 15 mA); the results were obtained at room temperature in the 2θ range from 2 to 100 s with a step of 0.02 and exposure time of 0.24 s at each point without rotation.

The Langmuir method was used for preparation of monolayers of compounds **1**–**5**, **8**, **10**, **12**. Monolayers of the ligands **1**–**5**, **8**, **10**, **12** were formed from their solutions in chloroform (0.02–0.1 mg/mL), which were spread onto the water subphase with a Hamilton microsyringe, at the air–water interface on a KSV NIMA Teflon trough equipped with a platinum Wilhelmy plate and two polyacetal barriers. The following conditions were employed for monolayer formation: The time of spreading solvent evaporation was 15 min, monolayer compression rate was 2.5 cm^2^/min, and t = 5–25 °C. KSV Nima/Attension 2.2 and Origin software packages were used for processing surface pressure/potential-molecular area isotherm data. Monolayers were vertically deposited onto quartz (Flyuorit, St.-Petersburg, Russia) or glass coated with ITO layer (Sigma-Aldrich, St. Louis, United States), which were degreased before use with ethyl alcohol and washed with deionized water.

UV-visible absorption spectra of solutions and Langmuir films of chromophores were recorded on a Perkin Elmer Lambda 35 spectrophotometer in the wavelength range of 200–1000 nm with a resolution of 1 nm. During the compression of monolayers on a Langmuir trough, UV/Vis differential reflection-absorption spectra of nitrocalixarene monolayers were recorded using a reflectometric probe connected to an AvaSpec-2048 fiber optic spectrophotometer. UV/Vis measurements were carried out at normal incidence [41].

The quantum chemical calculations of compounds **5′** and **2a’** were performed with the Gaussian 16 suit of programs [42]. Full geometry optimizations were carried out within the framework of the DFT (Becke 3-parameter Lee–Yang–Parr (B3LYP)) method using 6-31G(d) basis sets.

All the computations of compound **3** were performed using the Gaussian 16 suit of programs. The triple-ζ def-TZVP (TZVP stands for valence triple-zeta polarization) [43] basis set was used in all the cases. All the ground-state structures were optimized at DFT with the use of B3LYP functional [44,45]. The D3 London dispersion correction was applied as implemented in Gaussian [46]. To simulate the electronic spectra the time-dependent DFT was used [47,48,49]. All the simulated spectra were obtained by calculating the first 50 vertical excitations from the ground state (S_0_) equilibrium geometries with the use of cam-B3LYP long-range-corrected functional, which yield good results for charge-transfer systems [50,51,52]. Vertical transitions were broadened with Gaussian function of full width at half maximum of 0.24 eV. The dipole length formalism was used to calculate the oscillator strengths discussed in the present paper. For cam-B3LYP calculations all the excitation energies were consistently red-shifted by 0.25 eV in order to better match experimental spectral curves.

The X-ray diffraction data for the single crystal of **2a** were collected on a Bruker Kappa Apex II CCD diffractometer using graphite monochromated Mo*Kα* (0.71073 Å) radiation at 100(2) K. The performance mode of the sealed X-ray tube was 50 kV, 30 mA. The diffractometer was equipped with an Oxford Cryostream LT device. The suitable crystal of appropriate dimensions was mounted on a glass fiber in a random orientation. Preliminary unit cell parameters were determined with three sets of a total of 12 narrow frame scans. The data were collected according to a recommended strategy in an ω-scan mode. For data collection, images were indexed and integrated using the APEX3 data reduction package (v2019.1-0, Bruker AXS, Karlsruhe, Germany). Final cell constants were determined using global refinement of reflections from the complete dataset. Analysis of the integrated data did not show any decay. Data were corrected for systematic errors and absorption using SADABS-2016/2 [53], numerical absorption correction based on integration over a multifaceted crystal model and empirical absorption correction based on spherical harmonics according to the point group symmetry using equivalent reflections. XPREP-2014/2 and the ASSIGN SPACEGROUP routine of WinGX were used for analysis of systematic absences and space group determination. The structure was solved by the direct methods using SHELXT-2018/2 [54] and refined by the full-matrix least-squares on *F*^2^ using SHELXL-2018/3 [55]. Calculations were mainly performed using the WinGX-2018.3 suite of programs [56]. The position of hydrogen atom H1A of hydroxy group was determined by difference Fourier maps, and this atom was refined isotropically. The positions of the hydrogen atoms of the methyl group were found using a rotating group refinement with idealized tetrahedral angles. The other hydrogen atoms were inserted at the calculated positions and refined as riding atoms. Compound **2a** crystallized with half of the molecule in the asymmetric cell (*Z’* = 0.5). The compound studied had no unusual bond lengths and angles.

### Crystallographic Data for ***2a***

C_48_H_60_N_4_O_12_S_4_, colorless prism (0.522 × 0.331 × 0.319 mm^3^), formula weight 1013.24; triclinic, *P*$\bar{1}$ (No. 2), *a* = 10.3460(16) Å, *b* = 11.4906(18) Å, *c* = 12.590(2) Å, α = 99.893(3)°, β = 108.714(4)°, γ = 112.495(3)°, *V* = 1233.3(3) Å^3^, *Z* = 1, *Z’* = 0.5, T = 100(2) K, *d_calc_* = 1.364 g cm^−3^, μ(Mo*K*α) = 0.258 mm^−1^, *F*(000) = 536; *T*_max/min_ = 0.7749/0.6911; 9178 reflections were collected (2.225° ≤ θ ≤ 26.902°, index ranges: −13 ≤ *h* ≤ 9, −14 ≤ *k* ≤ 13, −14 ≤ *l* ≤ 15), 5134 of which were unique, *R_int_* = 0.0369, *R*_σ_ = 0.0606; completeness to θ of 25.242° 97.5%. The refinement of 312 parameters with no restraints converged to *R*_1_ = 0.0444 and *wR*_2_ = 0.0952 for 3709 reflections with *I* > 2σ(*I*) and *R*_1_ = 0.0713 and *wR*_2_ = 0.1091 for all data with *S* = 1.040 and residual electron density, ρ_max/min_ = 0.382 and −0.339 *e* Å^−3^. The crystals were grown by slow evaporation of a chloroform solution at r.t.

Deposition Number CCDC 2,024,073 contains the supplementary crystallographic data for this paper. These data are provided free of charge by the joint Cambridge Crystallographic Data Centre and Fachinformationszentrum Karlsruhe Access Structures service www.ccdc.cam.ac.uk/structures.

Procedures of synthesis and characterization data are as follows:

1^5^,3^5^,5^5^,7^5^-tetranitro-3^2^,7^2^-(dodecyloxy)-2,4,6,8-tetrathia-1,3,5,7(1,3)-tetrabenzenacyclooctaphane-1^2^,5^2^-diol (**2a**)

To a suspension of 1.00 g (1.48 mmol) of compound **1**, 0.97 g (3.69 mmol) of TPP, and 2.75 g (14.80 mmol) of C_12_H_25_OH in 30 mL of toluene, 0.58 mL (3.69 mmol) of DEAD was added dropwise and the reaction mixture was stirred for 8 h, at 25 °C, and then for 40 h, at 40 °C. The solvent was removed under reduced pressure and methanol (50 mL) was added to the residue; the precipitate was filtered. Yield 0.86 g (69%). T_m_ = 173 °C. R_f_ (CHCl_3_:MeOH = 100:1) 0.68. ^1^H NMR (CDCl_3_): 8.50 (s, 4H, H_7_), 8.49 (s, 4H, H_3_), 8.24 (s, 2H, OH), 4.05 (t, 4H, *J* 6.8, H_9_), 1.43 (q, 4H, *J* 7.2, H_10_), 1.26 (m, 20H, H_11–16_), 1.14 (m, 4H, H_17_), 1.04 (m, 4H, H_18_), 0.95 (m, 4H, H_19_), 0.88 (t, 6H, *J* 6.8, H_20_). ^13^C NMR (CDCl_3_): 163.9 (C_1i_), 161.9 (C_5i_), 144.2 (C_4i_), 140.5 (C_8i_), 131.1 (C_3_), 129.3 (C_7_), 129.2 (C_6i_), 120.8 (C_2i_), 77.0 (C_9_), 32.0 (C_10_), 29.7 (C_11_), 29.7 (C_12_), 29.6 (C_13_), 29.5 (C_14_), 29.5 (C_15_), 29.3 (C_16_), 29.3 (C_17_), 25.4 (C_18_), 22.8 (C_19_), 14.2 (C_20_). *m/z* (HRMS MALDI) (%) 1011.3036 (100) [M−H]^−^. Calculated for C_48_H_60_N_4_O_12_S_4_: 1011.3007 [M−H]^−^. IR (KBr, ν/cm^−1^) 3331 (OH), 3074, 2923 (C–H), 1522, 1339 (NO_2_). UV (CHCl_3_, 10^−5^ M (ε, 10^3^ M^−1^ cm^−1^)) 277.15 nm (66).

1^5^,3^5^,5^5^,7^5^-tetranitro-3^2^,7^2^-(octyloxy)-2,4,6,8-tetrathia-1,3,5,7(1,3)-tetrabenzenacyclooctaphane-1^2^,5^2^-diol (**2b**) 

To a suspension of 0.50 g (0.74 mmol) of compound **1**, 0.43 g (1.63 mmol) of TPP, and 1.20 mL (7.40 mmol) of C_8_H_17_OH in 30 mL of toluene, 0.25 mL (1.63 mmol) of DEAD was added dropwise and the reaction mixture was stirred for 40 h, at 40 °C. The solvent was removed under reduced pressure and ethanol (20 mL) was added to the residue; the precipitate was filtered. Yield 0.46 g (69%). T_m_ = 199 °C. R_f_ (CHCl_3_:MeOH = 100:1) 0.61. ^1^H NMR (CDCl_3_): 8.51 (s, 4H, H_7_), 8.49 (s, 4H, H_3_), 8.24 (s, 2H, OH), 4.05 (t, 4H, *J* 6.8, H_9_), 1.43 (q, 4H, *J* 7.2, H_10_), 1.22 (m, 8H, H_11,12_), 1.12 (m, 4H, H_13_), 1.04 (m, 4H, H_14_), 0.95 (m, 4H, H_15_), 0.86 (t, 6H, *J* 7.2, H_16_). ^13^C NMR (CDCl_3_): 163.8 (C_1i_), 161.9 (C_5i_), 144.1 (C_4i_), 140.4 (C_8i_), 132.2 (C_6i_), 131.0 (C_3_), 129.4 (C_7_), 129.1 (C_2i_), 77.0 (C_9_), 31.7 (C_10_), 29.3 (C_11_), 29.2 (C_12_), 29.2 (C_13_), 25.3 (C_14_), 22.7 (C_15_), 14.2 (C_16_). *m/**z* (HRMS MALDI) (%) 899.1730 (100) [M−H]^−^. Calculated for C_40_H_44_N_4_O_12_S_4_: 899.1755 [M−H]^−^. IR (KBr, ν/cm^−1^) 3329 (OH), 3077, 2926 (C–H), 1522, 1339 (NO_2_). UV (CHCl_3_, 10^−4^ M (ε, 10^3^ M^−1^ cm^−1^)) 274.53 nm (115).

7^2^-(dodecyloxy)-1^5^,3^5^,5^5^,7^5^-tetranitro-2,4,6,8-tetrathia-1,3,5,7(1,3)-tetrabenzenacyclooctaphane-1^2^,3^2^,5^2^-triol (**3**) 

To a suspension of 0.50 g (0.74 mmol) of compound **1**, 1.38 g (7.40 mmol) of C_12_H_25_OH, and 9.48 g (1.85 mmol) of TPP in 30 mL of toluene, 0.29 mL (1.85 mmol) of DEAD was added dropwise. After 5 days of stirring at 25 °C, the solvent was distilled under reduced pressure and methanol (10 mL) was added to the residue; the precipitate was filtered and purified by column chromatography (chloroform/methanol). Yield 0.050 g (7%). T_m_ = 224 °C (decomp.). R_f_ (CHCl_3_:MeOH = 20:1) 0.73. ^1^H NMR (CDCl_3_): 8.66 (dd, 4H, *J* 2.4, 4.4, H_7_), 8.44 (s, 2H, H_3_), 8.39 (s, 2H, H_11_), 4.44 (t, 2H, *J* 6.8, H_13_), 2.08 (q, 2H, *J* 7.2, H_14_), 1.58 (m, 2H, H_15_), 1.45 (m, 2H, H_16_), 1.26 (m, 14H, H_17–23_), 0.88 (t, 3H, *J* 6.8, H_24_). ^13^C NMR (CDCl_3_): 164.8 (C_9i_), 164.3 (C_1i_), 163.6 (C_5i_), 140.5 (C_4i_), 140.2 (C_8i,12i_), 134.3 (C_11_), 133.9 (C_7_), 133.6 (C_7′_), 133.1 (C_3_), 132.8 (C_2i_), 121.5 (C_6i_), 121.3 (C_6i’_), 120.4 (C_10i_), 79.9 (C_13_), 31.9 (C_14_), 29.7 (C_15_), 29.7 (C_16,17_), 29.6 (C_18_), 29.5 (C_19_), 29.4 (C_20_), 29.3 (C_21_), 25.6 (C_22_), 22.7 (C_23_), 14.1 (C_24_). *m/z* (HRMS MALDI) (%) 843.1103 (100) [M−H]^−^. Calculated for C_36_H_36_N_4_O_12_S_4_: 843.1129 [M−H]^−^. IR (KBr, ν/cm^−1^) 3406, 3240 (OH), 3074, 2924 (C–H), 1518, 1341 (NO_2_). UV (CHCl_3_, 10^−4^ M (ε, 10^3^ M^−1^ cm^−1^)) 272.57 nm (20).

3^2^,7^2^-bis(dodecyloxy)-1^5^,3^5^,5^5^,7^5^-tetrakis((*E*)-(4-nitrophenyl)diazenyl)-2,4,6,8-tetrathia-1,3,5,7(1,3)-tetrabenzenacyclooctaphane-1^2^,5^2^-diol (**5**)

To a suspension of 1.00 g (0.91 mmol) of compound **4**, 0.60 g (2.29 mmol) of TPP, and 1.70 g (9.10 mmol) of C_12_H_25_OH in 30 mL of toluene, 0.36 mL (2.29 mmol) of DEAD was added dropwise and the reaction mixture was stirred at 25 °C, for 8 h, and then for 8 h, at 40 °C. The solvent was distilled under reduced pressure and methanol was added to the residue and the precipitate was filtered. Yield 1.02 g (78%). T_m_ = 109 °C. R_f_ (CHCl_3_) 0.27. ^1^H NMR (CDCl_3_): 8.46 (s, 4H, H_11_), 8.43 (d, *J* 8.8, 2H, H_15_), 8.20 (s, 2H, OH), 8.08 (d, *J* 8.8, 2H, H_7_), 8.01 (d, *J* 8.8, 2H, H_14_), 7.70 (s, 4H, H_3_), 7.57 (d, *J* 8.8, 2H, H_6_), 4.51 (t, 4H, *J* 6.8, H_17_), 2.11 (q, 4H, *J* 7.2, H_18_), 1.65 (m, 4H, H_19_), 1.48 (m, 4H, H_20_), 1.38 (m, 4H, H_21_), 1.26 (m, 24H, H_22–27_), 0.87 (t, 6H, *J* 6.8, H_28_). ^13^C NMR (CDCl_3_): 162.1 (C_1i_), 161.3 (C_9i_), 155.7 (C_5i_), 154.7 (C_13i_), 149.1 (C_8i_), 148.9 (C_16i_), 148.8 (C_4i_), 145.5 (C_12i_), 132.2 (C_3_), 131.0 (C_11_), 130.3 (C_2i_), 125.0 (C_7_), 124.4 (C_15_), 123.5 (C_10i_), 123.4 (C_6_), 123.3 (C_14_), 78.1 (C_17_), 32.1 (C_18_), 30.1 (C_19_), 29.8 (C_20,21_), 29.8 (C_22_), 29.7 (C_23_), 29.5 (C_24_), 29.5 (C_25_), 25.9 (C_26_), 22.8 (C_27_), 14.2 (C_28_). *m/z* (HRMS MALDI) (%) 1451.4513 (100) [M+Na]^+^. Calculated for C_72_H_76_N_12_O_12_S_4_: 1451.4481 [M+Na]^+^. IR (KBr, ν/cm^−1^) 3358 (OH), 2923 (C–H), 1608 (N=N), 1525 (NO_2_), 1343 (NO_2_). UV (CHCl_3_, 10^−5^ M (ε, M^−1^ cm^−1^)) 353.93 nm (61633).

3^2^,7^2^-bis(dodecyloxy)-2,4,6,8-tetrathia-1,3,5,7(1,3)-tetrabenzenacyclooctaphane-1^2^,5^2^-diol (**7**)

To a suspension of 2.00 g (4.03 mmol) of thiacalix[4]arene 6, 2.64 g (10.07 mmol) of TPP, and 9.20 mL (40.30 mmol) of C12H25OH in 120 mL of toluene, 1.75 mL (10.07 mmol) of DEAD was added dropwise at 0 °C, and the reaction mixture was stirred at 25 °C, for 3 days. The solvent was distilled under reduced pressure and methanol (50 mL) was added to the residue; the precipitate was filtered. Yield 1.98 g (65%). T_m_ = 94 °C. R_f_ (Hex:EtOAc = 4:1) 0.69. ^1^H NMR (CDCl_3_): 7.62 (d, 4H, *J* 7.6, H_7_), 7.35 (s, 2H, OH), 6.83 (d, 4H, *J* 7.6, H_3_), 6.81 (t, 2H, *J* 7.6, H_8_), 6.48 (t, 2H, *J* 7.6, H_4_), 4.36 (t, 4H, *J* 6.8, H_9_) 1.98 (q, 4H, *J* 7.2, H_10_) 1.56 (m, 4H, H_11_), 1.41 (m, 4H, H_12_), 1.29 (m, 4H, H_13_), 1.25 (m, 24H, H_14–19_), 0.88 (t, 6H, *J* 6.8, H_20_). ^13^C NMR (CDCl_3_) 158.7 (C_1i_), 157.9 (C_5i_), 136.6 (C_3_), 134.7 (C_7_), 129.9 (C_6i_), 125.1 (C_4_), 123.1 (C_2i_), 119.5 (C_8_), 76.8 (C_9_), 32.0 (C_10_), 30.1 (C_11_), 29.6 (C_12_), 29.6 (C_13_), 29.6 (C_14_), 29.6 (C_15_), 29.6 (C_16_), 29.4 (C_17_), 26.0 (C_18_), 22.8 (C_19_), 14.2 (C_20_). *m/**z* (HRMS MALDI) (%) 965.2708 (100) [M+Cs]^+^. Calculated for C_48_H_64_O_4_S_4_: 965.2736 [M+Cs]^+^. IR (KBr, ν/cm^−1^) 3339 (OH), 2920 (C–H), 1456 (C_Ar_–C_Ar_).

3^2^,5^2^,7^2^-tris(dodecyloxy)-2,4,6,8-tetrathia-1,3,5,7(1,3)-tetrabenzenacyclooctaphane-1^2^-ol (**7′**)

To a suspension of 2.00 g (4.03 mmol) of thiacalix[4]arene 6, 2.64 g (10.07 mmol) of TPP, and 9.20 mL (40.30 mmol) of C_12_H_25_OH in 120 mL of toluene, 1.75 mL (10.07 mmol) of DEAD was added dropwise at 0 °C and the reaction mixture was stirred at 40 °C, for 10 h. The solvent was distilled under reduced pressure and methanol (50 mL) was added to the residue; the precipitate was filtered and purified by column chromatography using hexane/toluene = 4:1. Yield 0.75 g (23%). T_m_ = 73 °C. R_f_ (Hex:EtOAc = 4:1) 0.90. ^1^H NMR (CDCl_3_): 7.61 (d, 2H, *J* 7.6, H_3_), 7.58 (d, 2H, *J* 7.6, H_7_), 6.99 (t, 1H, *J* 7.6, H_4_), 6.83 (t, 1H, *J* 7.6, H_8_), 6.50 (m, 4H, H_11_), 6.36 (t, 2H, *J* 7.6, H_12_), 6.27 (s, 1H, OH), 4.17 (t, 2H, *J* 7.6, H_13_), 3.96 (t, 4H, *J* 6.8, H_25_), 1.98 (q, 2H, *J* 7.8, H_14_), 1.88 (q, 4H, *J* 7.6, H_26_), 1.57 (q, 4H, *J* 7.6, H_27_), 1.28 (br s, 50H, H_15–23,28–35_), 0.90 (br s, 9H, C_36,24_). ^13^C NMR (CDCl_3_) 161.3 (C_9i_), 158.4 (C_1i_), 157.1 (C_5i_), 136.3 (C_3_), 136.2 (C_11_), 133.4 (C_11′_), 132.9 (C_10i’_), 131.8 (C_7_), 131.6 (C_6i_), 130.0 (C_10i_), 124.6 (C_4_), 123.7 (C_2i_), 122.9 (C_8_), 119.4 (C_12_), 77.2 (C_25_), 74.2 (C_13_), 32.1 (C_26_), 32.1 (C_14_), 30.3 (C_27_), 30.1 (C_15_), 29.9 (C_16–19,28–31_), 29.9 (C_20_), 29.8 (C_32_), 29.7 (C_21_), 29.6 (C_33_), 29.5 (C_22_), 26.3 (C_34_), 26.1 (C_23_), 22.8 (C_35_), 14.2 (C_24,36_). *m/**z* (HRMS MALDI) (%) 1133.4606 (100) [M+Cs]^+^. Calculated for C_60_H_88_O_4_S_4_: 1133.4614 [M+Cs]^+^. IR (KBr, ʋ/cm^−1^) 3467 (OH), 2918 (C–H), 1430 (C_Ar_–C_Ar_).

3^2^,7^2^-bis(dodecyloxy)-1^5^,5^5^-dinitro-2,4,6,8-tetrathia-1,3,5,7(1,3)-tetrabenzenacyclooctaphane-1^2^,5^2^-diol (**8**) 

To a solution of 0.31 g (0.31 mmol) of compound **7** in 30 mL of chloroform, 0.39 g (1.10 mmol) of ${\mathrm{NO}}^{+}{\mathrm{NO}}_{3}^{–}$*18-crown-6 complex was added portionwise upon cooling to 0 °C, and the mixture was stirred for 12 h, at 10 °C. Then, water was added to the reaction mixture (10 mL) and organic phase was separated and dried over Na_2_SO_4_. After distillation of solvent, methanol (20 mL) was added to the residue and the precipitate was filtered. The product was isolated by column chromatography (CHCl_3_/MeOH = 100:1). Yield 0.080 g (24%). T_m_ = 151 °C. R_f_ (CHCl_3_:MeOH = 100:1) 0.71. ^1^H NMR (CDCl_3_): 8.64 (s, 2H, OH), 8.57 (s, 4H, H_3_), 7.17 (d, 4H, *J* 7.6, H_7_), 6.72 (t, 2H, *J* 7.6, H_8_), 4.33 (t, 4H, *J* 6.8, H_9_) 2.05 (q, 4H, *J* 7.6, H_10_) 1.60 (q, 4H, *J* 7.6, H_11_), 1.44 (q, 4H, *J* 7.6, H_12_), 1.25 (m, 14H, H_13–19_) 0.88 (t, 6H, *J* 7.2, H_20_). ^13^C NMR (CDCl_3_): 163.9 (C_1i_), 159.5 (C_5i_), 139.7 (C_4i_), 137.4 (C_8_), 132.2 (C_3_), 128.0 (C_7_), 126.0 (C_2i_), 123.2 (C_6i_), 78.2 (C_9_), 32.1 (C_10_), 29.9 (C_11_), 29.8 (C_12–15_), 29.7 (C_16_), 29.5 (C_17_), 25.8 (C_18_), 22.8 (C_19_), 14.3 (C_20_). *m/z* (HRMS MALDI) (%) 945.3305 (100) [M+Na]^+^. Calculated for C_48_H_62_N_2_O_8_S_4_: 945.3281 [M+Na]^+^. IR (KBr, ν/cm^−1^) 3303 (OH), 2917 (C–H), 1523 (NO_2_), 1341 (NO_2_). UV (CHCl_3_, 10^−4^ M (ε, 10^3^ M^−1^ cm^−1^)) 276.56 nm (7).

3^2^,5^2^,7^2^-tris(dodecyloxy)-1^5^-nitro-2,4,6,8-tetrathia-1,3,5,7(1,3)-tetrabenzenacyclooctaphane-1^2^-ol (**8′**) 

To a solution of 0.2 g (0.2 mmol) of compound **7′** in 30 mL of chloroform, 0.25 g (0.70 mmol) of ${\mathrm{NO}}^{+}{\mathrm{NO}}_{3}^{–}$*18-crown-6 complex was added portionwise upon cooling to 0 °C, and the mixture was stirred for 12 h, at 10 °C. Then, water was added to the reaction mixture (10 mL) and organic phase was separated and dried over Na_2_SO_4_. After distillation of solvent, methanol (20 mL) was added to the residue and the precipitate was filtered. The product was isolated by column chromatography (CHCl_3_/MeOH = 100:1). Yield 0.040 g (19%). ^1^H NMR (CDCl_3_): 8.47 (s, 2H, H_3_), 7.57 (d, 2H, *J* 7.6, H_7_), 7.34 (m, 4H, H_11_), 6.97 (t, 1H, *J* 7.6, H_8_), 6.42 (dd, 2H, *J* 7.6, H_12_), 4.11 (t, 2H, *J* 4.8, H_13_), 3.97 (br s, 4H, H_25_), 1.90 (t, 6H, H_14,26_), 1.54 (m, 4H, H_27_), 1.26 (br s, 50H, H_15–23,28–35_), 0.88 (br s, 9H, C_36,24_). *m/z* (ESI) (%) 1044.7 (100) [M−H]^−^. IR (KBr, ν/cm^−1^) 3310 (OH), 2915 (C–H), 1522 (NO_2_), 1341 (NO_2_).

1-nitro-4-(octyloxy)benzene (**10**) 

A suspension of 1.00 g (7.19 mmol) of *p*-nitrophenol **9** and 10.00 g (71.90 mmol) of K_2_CO_3_ in 10 mL of DMF was stirred for 1 h. at 80 °C. After addition of 1.86 mL (10.80 mmol) of 1-bromooctane, the reaction mixture was heated for 12 h. Then, the solvents were distilled under reduced pressure and water was added to the residue (50 mL) followed by extraction with 2 × 20 mL CH_2_Cl_2_; the organic phase was washed with 20 mL of water and 10 mL of brine and dried over Na_2_SO_4_. After filtration of Na_2_SO_4_, dichloromethane was distilled, and the oil product was purified by flash column chromatography (dichloromethane eluent). Yield 1.40 g (77%). T_m_ = 7 °C. R_f_ (CH_2_Cl_2_:Hex = 1:1) 0.60. ^1^H NMR (CDCl_3_): 8.18 (AB-d, *J* 9.2, 2H, H_3_), 6.93 (AB-d, *J* 9.2, 2H, H_2_), 4.04 (t, *J* 6.4, 2H, H_5_), 1.82 (q, *J* 7.4, 2H, H_6_), 1.46 (q, *J* 7.6, 2H, H_7_), 1.29 (m, 8H, H_8–11_), 0.89 (t, *J* 7.2, 3H, H_12_). ^13^C NMR (CDCl_3_): 164.4 (C_1_), 141.5 (C_4_), 126.0 (C_2_), 114.5 (C_3_), 69.1 (C_5_), 31.9 (C_6_), 29.4 (C_7_), 29.3 (C_8_), 29.1 (C_9_), 26.0 (C_10_), 22.8 (C_11_), 14.2 (C_12_). *m/**z* (HRMS MALDI) (%) 274.1418 (100) [M+Na]^+^. Calculated for C_14_H_21_NO_3_: 274.1414 [M+Na]^+^. IR (KBr, ν/cм^−1^) 2928 (C–H), 2856 (C–H), 1594 (C_Ar_–C_Ar_), 1514 (NO_2_), 1342 (NO_2_), 1112 (C–O). UV (CHCl_3_, 4 × 10^−5^ M (ε, 10^3^ M^−1^ cm^−1^)) 313.41 nm (14).

(*E*)-1-(4-nitrophenyl)-2-(4-(octyloxy)phenyl)diazene (**12**)

A suspension of 1.00 g (4.11 mmol) of (*E*)-4-((4-nitrophenyl)diazenyl)phenol **11** and 6.29 g (45.20 mmol) of K_2_CO_3_ in 10 mL of DMF was stirred for 1 h at 80 °C. After addition of 1.06 mL (6.17 mmol) of 1-bromooctane, the reaction mixture was heated for 12 h. Then, the solvents were distilled under reduced pressure, water was added to the residue (50 mL), the precipitate was filtered and washed threetimes with 20 mL of water and dried at 80 °C, for 4 h. Yield 1.38 g (95%). T_m_ = 65 °C. R_f_ (Hex:EtOAc = 4:1) 0.77. ^1^H NMR (CDCl_3_): 8.35 (d, *J* 9.2, 2H, H_3′_), 7.97 (d, *J* 6.0, 2H, H_2′_), 7.95 (d, *J* 6.4, 2H, H_3_), 7.02 (d, *J* 8.8, 2H, H_2_), 4.06 (t, *J* 6.4, 2H, H_5_), 1.83 (q, *J* 7.2, 2H, H_6_), 1.49 (q, *J* 7.6, 2H, H_7_), 1.30 (m, 8H, H_8–11_), 0.90 (t, *J* 6.8, 3H, H_12_). ^13^C NMR (CDCl_3_): 163.1 (C_1_), 156.2 (C_1′_), 148.4 (C_4′_), 147.0 (C_4_), 125.8 (C_2_), 124.8 (C_2′_), 123.2 (C_3′_), 115.1 (C_3_), 68.7 (C_5_), 31.9 (C_6_), 29.5 (C_7_), 29.3 (C_8_), 29.3 (C_9_), 26.1 (C_10_), 22.8 (C_11_), 14.2 (C_12_). *m/**z* (HRMS MALDI) (%) 356.1963 (100) [M+H]^+^. Calculated for C_20_H_25_N_3_O_3_: 356.1969 [M+H]^+^. IR (KBr, ν/cm^−1^) 2918 (C–H), 2852 (C–H), 1602 (N=N), 1580 (C_Ar_–C_Ar_), 1522 (NO_2_), 1340 (NO_2_), 1138 (C–O). UV (CHCl_3_, 5 × 10^−5^ M (ε, 10^3^ M^−1^ cm^−1^)) 379.39 nm (14).

## 4. Conclusions

A series of nitro- and 4-nitrophenylazo-thiacalixarenes partially substituted at lower and upper rims have been synthesized for the first time. Distally disubstituted tetranitrocalixarenes adopted *1,2-alternate* form in solution and solid state as confirmed by X-ray analysis and NMR spectroscopy. Nitro-containing calixarene chromophores displayed higher solid-state thermal stability than monomer phenols, whereas phenolic chromophores did not form rigid Langmuir monolayers at the air–water interface, and calixarenes demonstrated mixed behavior at the air–water interface. Parent nitrothiacalixarenes **1** and **4** formed bilayers with a low adhesion of compound **1** to ITO substrate after transfer. Long-chain lower rim derivatives, in contrast, formed true monolayers and were effectively transferred to ITO substrate. AFM visualization indicated more homogeneous film formation in the case of nitrothiacalixarene **3** with one dodecyl chain, with the highest collapse pressure at the air–water interface and a bathochromic shift of absorption band on going from solution to one monolayer. This bathochromic shift was rationalized by the change of the polarity of environment upon transition from chloroform solution to air–water interface and solid substrate, whereas the appearance of a shoulder band at 350 nm was attributed to aggregation of calixarene molecules and supported by ab initio studies and dynamic light scattering.

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
