# Peer review of "Thermally Stable Nitrothiacalixarene Chromophores: Conformational Study and Aggregation Behavior"

_ijms, 2020, doi:10.3390/ijms21186916_

Round 1
Reviewer 1 Report
Manuscript ID: ijms-932660
This is a nice piece of work illustrating the synthesis of nitrothiacalixarene chromophores and demonstrating their higher thermal stability and aggregation behaviour. The work was neatly executed and the observed results are properly interpreted with clear discussions. This work may be recommended for publication in IJMS with minor revision.
- Figure 1 resolution needs to be improved. It is recommended to use chemdraw to draw these structures.
- Manuscript has to be checked for typographical errors “Iteroselective nitration”
- Figure 8: It seems to be bit noisy. It can be recorded again to recheck it.
- The following references may be included in the manuscript in appropriate places (Polymer Journal, 2003, 35, 213-229; Polymer Degradation and Stability, 80, 2003, 203-208; Biomolecules, 2019 9, 90).
Author Response
Figure 1 resolution needs to be improved. It is recommended to use chemdraw to draw these structures.
Response 1. Resolution of Fig. 1 was improved.
Manuscript has to be checked for typographical errors “Iteroselective nitration”
Response 2. The word “Iteroselective” was replaced by “Selective” (line 108).
Figure 8: It seems to be bit noisy. It can be recorded again to recheck it.
Response 3. The UV/visible reflection–absorption spectrum (UVRAS) given as a main plot in Fig. 8 is recorded from a beam reflected from the monolayer of compound 3 at the air–water interface, in which there are ultralow quantities of compound (comparable to nano/micromolar bulk solutions). Under these conditions, the absorbance peak intensity is comparable to background intensity, which appears as noise in the spectrum and could not be further improved in terms of resolution. It can be seen in Figs. S41–S46 that UVRAS of all nitrocalixarene monolayers recorded upon monolayer compression look noisy compared to UV/Vis absorbance spectra of calixarene solutions in transmission mode. UVRAS is a well-known technique for in situ analysis of optical characteristics of monolayers before their transfer to solid substrates and is equally well applicable for spectral analysis of monolayers at low and high surface pressures [Adv. Colloid Interface Sci., 2015, 225, 134–135; New J. Chem., 2019, 43, 11419–11425].
The following references may be included in the manuscript in appropriate places (Polymer Journal, 2003, 35, 213-229; Polymer Degradation and Stability, 80, 2003, 203-208; Biomolecules, 2019 9, 90).
Response 3. Included the references [Polymer Journal, 2003, 35, 213-229] (ref. 13, lines 756–759) and [Biomolecules, 2019 9, 90] (ref. 7, lines 741–743) to the manuscript.

Reviewer 2 Report
The manuscript "Thermally stable nitrothiacalixarene chromophores: conformational study and aggregation behavior" by Muravev et al. reports a very detailed experimental characterization and the computational investigation. The manuscript is clear and complete. I think that the manuscript deserves publication after an improvement of the computational part with more clearer pictures of the optimized structures and by the introduction of a figure with the involved molecular orbitals in the electronic transitions. After these minor variations, the manuscript is able to be published.
Author Response
The manuscript "Thermally stable nitrothiacalixarene chromophores: conformational study and aggregation behavior" by Muravev et al. reports a very detailed experimental characterization and the computational investigation. The manuscript is clear and complete. I think that the manuscript deserves publication after an improvement of the computational part with more clearer pictures of the optimized structures and by the introduction of a figure with the involved molecular orbitals in the electronic transitions. After these minor variations, the manuscript is able to be published.
Response: The optimized structures in Fig. 11 were enlarged to the column width; side view of the dimers was left intact in order to visualize both calixarene conformation and mutual orientation of macrocyclic scaffolds. The table containing the figures with the involved molecular orbitals in the electronic transitions was added to the Supplementary (Table S2) and Figure S49 was thus deleted. There was a technical error in the description of UV absorption spectra of dimers a–e, with indication of intramolecular energy transfer only in dimer d. In fact, both lateral dimers d and e show such transitions. Corrected in the main text (lines 426–433) and clarified the statement in lines 442–443 (“aggregation with closely located nitro head groups from different molecules” instead of “most types of aggregation”)

Reviewer 3 Report
The article of Dr. Anton Muravev and co-authors entitled “Thermally stable nitrothiacalixarene chromophores: conformational study and aggregation behavior” (Manuscript ID: ijms-932660) does not meet the requirements of an article addressed to International Journal of Molecular Science.
The article is thematically quite up-to-date, unfortunately the authors generally omitted the current source literature, moreover, there is a large number of errors generally related to the lack of analytical samples, lack of 2D NMR spectroscopic analyzes (HMQC, HMBC, etc., generally at the aromatic region) and, above all, to serious discrepancies between the evidence of NMR spectra included in Supplementary File(s) and their description in the main part.
Before further steps, the material should be thoroughly reviewed in the experimental part related to the preparation of analytical samples and the tests carried out on them, and there are no expected applications in chapter 3, which was omitted in the published version.
If the editorial office is inclined to further stages of production, I list of the glaring shortcomings, some including other minor noticed errors, and encourage the authors to further work on this extremely promising subject: At line 26 is: … monoalkylated 3 … , but should be better: monoalkylated calixarene 3 … . At line 41 is: … as acceptor … , but did the authors mean? … as electron acceptor … . The works addressed to Int. J. Mol. Sci. are aimed at a wide range of readers, and you should also write information that is obvious to a narrow audience. At line 42 is: … as donor … , but did the authors mean? … as electron donor … . There is imprecise information on line 65 in Scheme 1 and on the line 81 in Figure 1 three times: … NO2PhN2 … , but should be ; … 4-NO2C6H4N2 … . The authors suggest that both the nitro (NO2) and azo (N2) groups are misplaced on the same aromatic carbon at benzene ring, making it a five-bond carbon. In line 65 of Scheme 1 there is another inaccurate information twice: … =any group … , but should be: … =H … . In line 65 of Scheme 1 there is another inaccurate information twice: … other group … . Please clarify and define these other groups. There is subsection 2.1. in the line 83, but subsection 2.1. is already on line 72! Please corrected carefully. Also at line 122 is subsection 2.2., but should be 2.3. Also at line 187 is subsection 2.3., but should be 2.4. Also at line 218 is subsection 2.3., but should be 2.5. Also at line 346 is subsection 2.4., but should be 2.6. Also at line 392 is subsection 2.5., but should be 2.7. In lines 457–458, the desired chapter 3 on the application proving the hypotheses is missing. Is the chapter number correct on line 458? Is the chapter number correct on line 694? In the lines 100–101 Scheme 2 requires redrafting to the style of Scheme 3 with an indication of the details in the form of the number of reagents necessary to obtain products 2a, 2b, 3 and 5. Reagents and details can be entered in the text of the caption under scheme 2, and the top left benzene ring in compounds 2a and 2b is too large. At scheme 3 at line 116 is … 10-25°C … , but should be better: … 10–25°C … . Please uses the mean sign ( – ) between the numbers generally. At Scheme 4 al line 120 there is no space between the comma and DMF. At lines 140–141 is: … shows two triplets at 6.97 (1H) and 6.42 ppm (2H)… , but should be: … shows triplet at 6.97 (1H) and doublet-of-doublets at 6.42 ppm (2H)… . One proton H8 generate one signal, and the H12 protons that the proton interacts with chemically different protons H14 and H14’ to give dd type of signals. Consistently at line 665 is … 6.42 (t, 2H … , but should be … 6.42 (dd, 2H … . Also the spin-spin constants coupling 3JH-H = 4.8 Hz at 6.97 ppm and 6.42 ppm of the arene ring there is no physical sense. Must be like 8 Hz. This is due to the lack of an analytical sample of the compound 8’. For example, in lines 222, 225 and 680 there is a mathematical sign of multiplication (∙) other than in line 236 ( × ) and in line 680 ( * ), respectively. Please standardize everywhere as ∙ or × , but nor *, such as in lines: 435, 467, 550 (twice), 693, etc. For example at lines 222 and 236 there are different mathematical signs of subtraction ( - or – ) - please standardize everywhere.
At line 467 is:… experiments … , but should be better: … spectra … . At the line 469 is the lack if the spin-spin constants coupling definition. At lines 489 and 501 there is a space character before the degree mark Celsius (°C), but there is no space character in other places, for example on lines 566, 579, 66-46, etc. Please standardize everywhere. At the line 658 is lack of the 13C-NMR and FT-IR spectra. Generally, until line 563 to line 696 The multiplicity and constants of spin-spin coupling should be thoroughly checked and revised.At line 691 is at IR: … 2918 (CAr–H) … , but the signal is in the aliphatic region. Please correct also in lines 679, 656, 628, 615, 600 (2924 signal), 586 (2926 signal), 574 (2923 signal).
For example, at line 597 is lacks of the C12 and 1i signals at the 13C-NMR spectrum of compound 3. In the chapter References at lines 728–870, after the written year, the authors use a comma ( , ), written arbitrarily, sometimes in bold ( , ), other times in normal font. See lines 842 and 833 for an example. Please standardize. On lines 749, 751 and 821, the prefix p (para) should rather be in italics. See line 768. On line 821, the tert should rather be in italics.
Author Response
The article is thematically quite up-to-date, unfortunately the authors generally omitted the current source literature, moreover, there is a large number of errors generally related to the lack of analytical samples, lack of 2D NMR spectroscopic analyzes (HMQC, HMBC, etc., generally at the aromatic region) and, above all, to serious discrepancies between the evidence of NMR spectra included in Supplementary File(s) and their description in the main part.
1) Response: 2D NOESY NMR spectroscopy analysis was applied to target nitrothiacalixarene chromophores 3 and 8 and clear statements of stereoisomeric form were made; 2D NMR techniques are obviously unnecessary in the case of model compounds 10 and 12 and NOESY NMR is impractical in case of nitrocalixarenes 2a,b due to close chemical shifts of the signals of aromatic protons 3 and 7 (their cross-peak could not be distinguished from the diagonal peaks). Partial 2D NOESY spectra indicating key NOESY cross-peaks were added as insets to supplementary 1H NMR spectra of nitrothiacalixarenes 3 (Fig. S9, 7/3,11 cross-peaks) and 8 (Fig. S23, 3/7 cross-peak). The authors apologize for discrepancies of supplementary NMR spectra and their description in the main part, which were caused by the technical error; detailed response is given in 17) Response.
Before further steps, the material should be thoroughly reviewed in the experimental part related to the preparation of analytical samples and the tests carried out on them, and there are no expected applications in chapter 3, which was omitted in the published version.
2) Response: Chapter 3 was not written due to the extensive discussion of the results in chapter 2 including expected applications (explosives in section 2.4, lines 202–205; nonlinear optical materials in section 2.7, lines 452–460).
If the editorial office is inclined to further stages of production, I list of the glaring shortcomings, some including other minor noticed errors, and encourage the authors to further work on this extremely promising subject: At line 26 is: … monoalkylated 3 … , but should be better: monoalkylated calixarene 3 … .
3) Response: Corrected into “monoalkylated calixarene 3” (lines 26,27).
At line 41 is: … as acceptor … , but did the authors mean? … as electron acceptor … . The works addressed to Int. J. Mol. Sci. are aimed at a wide range of readers, and you should also write information that is obvious to a narrow audience. At line 42 is: … as donor … , but did the authors mean? … as electron donor … .
4) Response: Corrected into “electron-withdrawing” (line 41) and “electron donor” (line 42), respectively.
There is imprecise information on line 65 in Scheme 1 and on the line 81 in Figure 1 three times: … NO2PhN2 … , but should be ; … 4-NO2C6H4N2 … .
5) Response: Corrected into “4-NO2C6H4N2” in Scheme 1 and Figure 1.
The authors suggest that both the nitro (NO2) and azo (N2) groups are misplaced on the same aromatic carbon at benzene ring, making it a five-bond carbon.
6) Response: Having replaced “NO2C6H4N2” by “4-NO2C6H4N2” in response to previous comment, it becomes obvious that nitro and azo groups are located at para-position to each other at benzene ring and abnormal valence state of any carbon atom in benzene ring is thus impossible.
In line 65 of Scheme 1 there is another inaccurate information twice: … =any group … , but should be: … =H ….
7) Response: Original information is correct, because Scheme 1 outlines general substitution pattern of nitrocalixarene chromophores known in literature, not only this work. “Any group” stands for either H or alkyl group.
In line 65 of Scheme 1 there is another inaccurate information twice: … other group … . Please clarify and define these other groups.
8) Response: Changed from “other group” to “any group except H”. This group could be alkyl group with various terminal substituents; there are ca. 200 references on nitrocalixarene derivatives, so the list of possible terminal substituents is too large to define them all.
There is subsection 2.1. in the line 83, but subsection 2.1. is already on line 72! Please corrected carefully. Also at line 122 is subsection 2.2., but should be 2.3. Also at line 187 is subsection 2.3., but should be 2.4. Also at line 218 is subsection 2.3., but should be 2.5. Also at line 346 is subsection 2.4., but should be 2.6. Also at line 392 is subsection 2.5., but should be 2.7.
9) Response: Corrected.
In lines 457–458, the desired chapter 3 on the application proving the hypotheses is missing. Is the chapter number correct on line 458? Is the chapter number correct on line 694?
10) Response: Chapter 3 was not written due to the extensive discussion of the results in chapter 2 including perspective applications. There is a strong link between description and discussion of the results and moving the text of discussion to a separate section disrupts the flow of main text. The Instructions for Authors of IJMS allow to combine the Discussion section with Results (https://www.mdpi.com/journal/ijms/instructions#suppmaterials –> Manuscript preparation –> Research Manuscript Sections –> Discussion). Corrected chapter numbers on lines 461 and 697 into chapters 3 and 4, respectively.
In the lines 100–101 Scheme 2 requires redrafting to the style of Scheme 3 with an indication of the details in the form of the number of reagents necessary to obtain products 2a, 2b, 3 and 5. Reagents and details can be entered in the text of the caption under scheme 2, and the top left benzene ring in compounds 2a and 2b is too large. At scheme 3 at line 116 is … 10-25°C … , but should be better: … 10–25°C … . Please uses the mean sign ( – ) between the numbers generally.
11) Response: Redrafted Scheme 2 style to that of Scheme 3. Added details of the amounts of reagents to the caption of Scheme 2. Reduced the size of top left benzene ring in compounds 2 to standard size.
At Scheme 4 al line 120 there is no space between the comma and DMF.
12) Response: Corrected.
At lines 140–141 is: … shows two triplets at 6.97 (1H) and 6.42 ppm (2H)… , but should be: … shows triplet at 6.97 (1H) and doublet-of-doublets at 6.42 ppm (2H)… . One proton H8 generate one signal, and the H12 protons that the proton interacts with chemically different protons H14 and H14’ to give dd type of signals. Consistently at line 665 is … 6.42 (t, 2H … , but should be … 6.42 (dd, 2H … . Also the spin-spin constants coupling 3JH-H = 4.8 Hz at 6.97 ppm and 6.42 ppm of the arene ring there is no physical sense. Must be like 8 Hz. This is due to the lack of an analytical sample of the compound 8’.
13) Response: Agree with this comment, made changes to the main text (line 146). Recalculated spin coupling constants and corrected in the text, the resulting constant is 7.6 Hz both at 6.97 and 6.42 ppm (line 668).
For example, in lines 222, 225 and 680 there is a mathematical sign of multiplication (∙) other than in line 236 ( × ) and in line 680 ( * ), respectively. Please standardize everywhere as ∙ or × , but nor *, such as in lines: 435, 467, 550 (twice), 693, etc. For example at lines 222 and 236 there are different mathematical signs of subtraction ( - or – ) - please standardize everywhere.
14) Response: Standardized into sign of multiplication (×) and mathematical sign of subtraction (–) everywhere.
At line 467 is:… experiments … , but should be better: … spectra … . At the line 469 is the lack if the spin-spin constants coupling definition.
15) Response: corrected into “spectra” (line 470). Added phrase “Spin-spin coupling constants are given in Hz” (line 472).
At lines 489 and 501 there is a space character before the degree mark Celsius (°C), but there is no space character in other places, for example on lines 566, 579, 66-46, etc. Please standardize everywhere.
15) Response: Standardized into °C everywhere.
At the line 658 is lack of the 13C-NMR and FT-IR spectra.
16) Response: 13C NMR spectrum of compound 8’ was not recorded due to its low solubility in deuterated solvent. Assignment of FT-IR bands was added (line 670).
Generally, until line 563 to line 696 The multiplicity and constants of spin-spin coupling should be thoroughly checked and revised.At line 691 is at IR: … 2918 (CAr–H) … , but the signal is in the aliphatic region. Please correct also in lines 679, 656, 628, 615, 600 (2924 signal), 586 (2926 signal), 574 (2923 signal).
17) Response: The authors apologize for the technical error related to contradiction of supplementary 1H NMR spectra and their description in text. The multiplicity and spin coupling constants were revised and corrected. The assignment of IR absorption bands at ca. 2920 cm–1 was changed to “(C–H)” (asymmetric stretching vibrations of C–H bonds of alkyl chain).
For example, at line 597 is lacks of the C12 and 1i signals at the 13C-NMR spectrum of compound 3.
18) Response: Added assignment of these carbon atoms to the experimental part (line 599).
In the chapter References at lines 728–870, after the written year, the authors use a comma ( , ), written arbitrarily, sometimes in bold ( , ), other times in normal font. See lines 842 and 833 for an example. Please standardize.
19) Response: Standardized.
On lines 749, 751 and 821, the prefix p (para) should rather be in italics. See line 768. On line 821, the tert should rather be in italics.
20) Response: Highlighted the prefixes in italics.
On behalf of the co-authors, I would like to thank Reviewer for careful inspection of the manuscript text and valuable comments that contributed to the improvement of appearance of the revised manuscript.

Round 2
Reviewer 3 Report
A thoroughly revised good article of Dr. Anton Muravev and co-authors entitled “Thermally stable nitrothiacalixarene chromophores: conformational study and aggregation behavior” (Manuscript ID: ijms-932660) meets the requirements of an article for the International Journal of Molecular Science.
The authors added one current reference (from 2019), moreover, the authors revised the technical information and introduced selected 2DNOESY experiments, which cleared up a number of misunderstandings, enabling a thorough revision of the description of spectroscopic data.
Listed below are some minor errors with regard to the revised material to be corrected in the following production steps:
In 5 response authors report that they have entered corrections "4-NO2C6H4N2" but the information is inaccurate. At Scheme 1 and at the Figure 1 is "4-NO2PhN2", but must be “4-NO2C6H4N2". Ph = C6H5 but not C6H4. Ph has one site to bind a substituent and there are two substituents. Please correct.
With reference to answers 7 and 8 in the caption under Scheme 1, which looks like a figure (analogously in the caption under Figure 1), it would be good to refer to at least one reference in which this method of writing is used.
The authors accidentally treat the concepts of spectrum and experiment / s. Spectroscopy (1H-NMR, 13C-NMR, FT-IR) provides physicochemical data, but a spectroscopic experiment/s (such DEPT-135 and NOESY) does not provide them, but only informs about correlations, interactions or the nature of the signals. Refer to SI for descriptions No: S2, S6, S10, S14, S18, S21, S23, S24, S30, S34, and correct.
The baseline in the 1H-NMR spectrum (fig. S29) needs to be corrected to a horizontal line over the entire range given.
Author Response
In 5 response authors report that they have entered corrections "4-NO2C6H4N2" but the information is inaccurate. At Scheme 1 and at the Figure 1 is "4-NO2PhN2", but must be “4-NO2C6H4N2". Ph = C6H5 but not C6H4. Ph has one site to bind a substituent and there are two substituents. Please correct.
Response: Replaced "4-NO2PhN2" by “4-NO2C6H4N2" in Scheme 1 and Figure 1.
With reference to answers 7 and 8 in the caption under Scheme 1, which looks like a figure (analogously in the caption under Figure 1), it would be good to refer to at least one reference in which this method of writing is used.
Response: Scheme 1 was changed into Figure 1 and subsequent figures and schemes were renumbered both in captions and in the manuscript text. The “[NO2]” was replaced by “R”, “any group” was replaced by “H or alkyl group or CH2X, where X is functional substituent (…)”; “any group except H” was replaced by “CH2X, where X is functional substituent (…)”. The references specifying substituents in Figure 1 (former Scheme 1) were added and also mentioned in main text (line 53); subsequent references were renumbered in the main text. With reference to comment on explanation of the method of writing, top view of calixarenes is often used, when conformation of molecules could not be unambiguously deduced or differs depending on lower-rim substitution pattern (for example, ref. [3] (chapters 2–4, 6–11, 13–17, 22, 24, 30, 31, 33, 34, 37, and 38) in the text or more recent review [Synthesis 2017, 49, 1009–1023].
The authors accidentally treat the concepts of spectrum and experiment / s. Spectroscopy (1H-NMR, 13C-NMR, FT-IR) provides physicochemical data, but a spectroscopic experiment/s (such DEPT-135 and NOESY) does not provide them, but only informs about correlations, interactions or the nature of the signals. Refer to SI for descriptions No: S2, S6, S10, S14, S18, S21, S23, S24, S30, S34, and correct.
Response: Replaced “13C and DEPT NMR spectra” by “13C NMR spectrum and DEPT-135 experiment” in Figures S2, S6, S10, S14, S18, S21, S24, S30, and S34; replaced “partial 2D NOESY spectrum” by “2D NOESY experiment” in Figures S23 and S9. DEPT-135 experiment was not carried out for compound 8 and is not shown in Fig. S24 and the caption was thus corrected.
The baseline in the 1H-NMR spectrum (fig. S29) needs to be corrected to a horizontal line over the entire range given.
Response: Phase correction was applied to the baseline in Figure S29 so that it appears as a horizontal line.
